# Risk and Protective Factors in Adolescent Suicidal Behaviour: A Network Analysis

**DOI:** 10.3390/ijerph19031784

**Published:** 2022-02-04

**Authors:** Eduardo Fonseca-Pedrero, Susana Al-Halabí, Alicia Pérez-Albéniz, Martin Debbané

**Affiliations:** 1Department of Educational Sciences, University of La Rioja, 26004 Logrono, Spain; alicia.perez@unirioja.es; 2Department of Psychology, University of Oviedo, 33003 Oviedo, Spain; alsusana@uniovi.es; 3Faculty of Psychology and Educational Sciences, University of Geneva, 1205 Geneva, Switzerland; martin.debbane@unige.ch; 4Department of Clinical, Educational and Health Psychology, University College London, London WC1E 6BT, UK

**Keywords:** suicidal behaviour, network, risk factors, protective factors, adolescents

## Abstract

Given that death by suicide continues to rank among the top three causes of death during adolescence, new psychological models may contribute critical insight towards understanding the complex interactions between risk and protective factors in suicidal behaviour. The main objective of this study was to analyse the psychological network structure of suicidal behaviour and putative risk and protective factors in school-aged adolescents. Methods: Stratified random cluster sampling was performed. The final sample comprised 1790 students (53.7% female, *M* = 15.7 years, *SD* = 1.26). Instruments were administered to assess suicidal behaviour, emotional and behavioural difficulties, prosocial behaviour, subjective well-being, self-esteem, depressive symptomatology, academic performance, socio-economic status, school engagement, bullying, and cyberbullying. Results: In the estimated psychological network, the node with the highest strength was depressive symptomatology, and that with the highest expected influence value was bullying. Suicidal behaviour was positively connected to symptoms of depression and behavioural problems. In addition, suicidal behaviour was negatively connected to self-esteem and personal well-being. The results of the stability analysis indicated that the network was accurately estimated. Conclusions: Suicidal behaviour can be conceptualised as a dynamic, complex system of cognitive, emotional, and affective characteristics. New psychological models allow us to analyse and understand human behaviour from a new perspective, suggesting new forms of conceptualisation, evaluation, intervention, and prevention.

## 1. Introduction

Suicide is a global health problem [1] that is the leading non-natural cause of death in adolescents and young adults worldwide [2,3]. It is also one of the main causes of premature death and years of disability in life. The need to address suicidal behaviour in adolescence comes from more than just the data on prevalence during this developmental stage. There are several reasons, the more important of which include: the fact that suicidal behaviours and completed suicides in the child–adolescent population have increased in recent decades [4], more suicides committed at younger ages are being recorded [5], most people who have considered or attempted suicide did so for the first time in their youth, typically before the age of twenty [6], suicidal ideation is a well-established predictor of new suicide attempts in the future and of substantial problems for young peoples’ social and emotional development beyond adolescence, as well as a risk factor for completed suicides [7,8], and the emotional impact that the suicide of a minor has on the family and on society is quite considerable. It is a real family tragedy compounded by social stigma.

Suicidal behaviour is a complex, multidimensional, multifactorial phenomenon, associated with stigma and taboos [9,10]. Its conceptual delimitation, aetiology, assessment, treatment, and prevention is a complex task with no simple solutions [11,12,13]. Currently, many questions related to suicidal behaviour are still unresolved [14]. One is related to a lack of conceptual definition or few validated theoretical psychological models, an aspect that also affects assessment, intervention, prevention, and postvention. It would therefore be interesting to enrichen the theoretical model by data-driven models such as those emerging in different fields, such as neuroscience, public health, etc., that allow the conceptualisation and understanding of suicidal behaviour from different perspectives [15].

The prevalence of suicidal behaviours during adolescence is high, as are the associated costs on a personal, family, social, academic, and socio-economic level. For instance, a meta-analysis conducted by Lim et al. [16] found that life prevalence and 12-month prevalence for suicide attempts in adolescents were 6% (95% CI: 4.7–7.7%) and 4.5% (95% CI: 3.4–5.9%), respectively. In addition, for suicidal ideation the life prevalence and 12-month prevalence were 18% (95% CI: 14.2–22.7%) and 14.2% (95% CI: 11.6–17.3%), respectively. Young people who present suicidal behaviour (e.g., ideation, planning, attempts) also report, amongst others, more emotional and behavioural problems, higher substance use, more risky behaviours and impulsivity, and poorer quality of life, self-esteem, and emotional regulation [5,10,17,18,19,20]. Furthermore, suicidal behaviour has been associated with a wide amalgam of risk and protective factors [6,10,21,22,23,24,25]. Risk factors include the presence of a mental disorder, previous attempts, psychological factors (e.g., hopelessness, impulsivity, cognitive rigidity), family history of mental disorders or previous attempts, bullying, cyberbullying, and trauma, to name a few. Protective factors are less well-studied, and include but are not limited to problem-solving ability, social-emotional skills, limited access to the means of suicide, cultural and religious beliefs that discourage suicide, and social and family support [3]. There is no doubt that appropriate understanding, evaluation, and intervention in suicidal behaviour requires the analysis of both the phenomenon itself (e.g., frequency, intensity, duration) and the associated risk and protective factors.

Several models of suicide have been developed and validated in recent years [9,10,23]. Theoretical models of suicide behaviours are important to understand and prevent this complex and multifactorial phenomenon of human behaviour. The network model has emerged with new strength in psycho(patho)logy as a response to, among other things, some of the problems associated with the biomedical model of “common latent cause” (e.g., reification, tautological reasoning, categorical nature) [26,27]. It is plausible that this way of conceptualizing mental disorders and human behaviour is one of the main obstacles, although not the only one, preventing progress in this scientific field. For this reason, many voices advocate for a radical paradigm shift and a profound re-conceptualisation of classifying systems, with the network model being one of the possible solutions [28,29]. The network model considers psychological problems as a complex system of symptoms (or signs, traits, mental states, phenomena, etc.) that causally impact or interact with each other. Therefore, an underlying latent variable would not be the common cause of the covariance between symptoms. Based on this approach, psychological problems would vary as a result of differences in the number, nature, and interrelatedness of the elements [30] within (psychological level) and across levels (bio-psycho-social) and time. In addition, this approach is presented as a new perspective from which to analyse and understand psychological phenomena such as suicidal behaviour [31,32]. In the suicide arena, the strength of the network model is to characterise the nature of the dynamics between variables around a target behaviour (i.e., planning), that is susceptible to occur through a variety of interactions. In essence, this approach may allow a more detailed appreciation of suicidal behaviour and, therefore, could usefully contribute to the refinement of existing explanatory models in this field and to the establishment of new therapeutic targets and prevention strategies, among others [33,34].

To date, however, the network model has not been used in the analysis of suicidal behaviour and its relationship with various putative risk and protective factors in a representative sample of adolescents [35]. The network approach is perhaps the only model that can yield valuable information when mixing together behavioural, psychological, and environmental data. In addition, adolescence represents a critical time window of opportunity for the prevention of suicidal behaviour. We must therefore continue to identify risk and protective factors for suicidal behaviour in this developmental stage [36]. Within this research framework, the main objective of this study was to analyse the psychological network structure of suicidal behaviour and the various putative risk and protective factors (emotional and behavioural difficulties, prosocial behaviour, subjective well-being, self-esteem, depressive symptomatology, academic performance, socio-economic status, school engagement, bullying, and cyberbullying). Estimators of protective factors (well-being, prosocial behaviour, self-esteem, school engagement) are expected to be negatively related to suicidal behaviour, while risk factors such as emotional and behavioural problems (hyperactivity, emotional problems, depressive symptoms, etc.) are expected to be positively related to suicidal behaviour.

## 2. Materials and Methods

### 2.1. Participants

Stratified random cluster sampling was performed, with the classroom as the sampling unit, from a population of 15,000 students in the region of La Rioja (northern Spain). Strata was created as a function of the geographical zone and the current stage in the educational cycle. An initial sample comprised 1972 students. Students with more than 2 points (*n* = 146) on the Oviedo Infrequency Scale–Revised or who were over 19 years old (*n* = 36) were eliminated.

The final sample comprised 1790 students, 816 (45.6%) were male, 961 (53.7%) were female, and 13 participants (0.7%) reported another gender identity. The mean age was 15.7 years (*SD* = 1.26; age range = 14 to 18). Distribution by age was as follows: 14 years, *n* = 342; 15 years, *n* = 541; 16 years, *n* = 410; 17 years, *n* = 299; 18 years, *n* = 198.

Most of the participants (89.9%) were Spanish, 2.5% were Romanian, 1.8% Latin American, 1.4% Moroccan, 0.8% Pakistani, 0.3% Portuguese, and 3.8% were from other nationalities.

### 2.2. Instruments

In this research, we used self-reports that yielded information on different levels (behaviour, well-being, individual psychological factors, interpersonal dimensions, psychopathological symptoms, perceived environmental characteristics) and that are linked to suicide research [5,24,25].

The Paykel Suicide Scale (PSS) [37]. The PSS is a self-report tool designed for the evaluation of suicidal behaviour. The tool consists of 5 items with a dichotomous (yes/no) response. The scores range from 0 to 5. The Spanish adaptation of the PSS has demonstrated adequate psychometric properties [38].

The Personal Well-being Index–School Children (PWI-SC) [39]. This index has 8 items, with response options ranging from 0 (completely dissatisfied) to 10 (completely satisfied). The PWI-SC items assess subjective satisfaction with a specific area of life in a relatively generic, abstract way. The first item on the scale, analysing “life as a whole”, was used in the present study. The validated Spanish version of the PWI-SC was used in the present study, where Cronbach’s alpha for the total score was 0.83 [40].

The Strengths and Difficulties Questionnaire (SDQ) [41]. The SDQ is a self-report questionnaire that is widely used for the assessment of different emotional and behavioural difficulties related to mental health in adolescents. The SDQ is made up of 25 statements in 5 subscales: emotional symptoms, conduct problems, hyperactivity, peer problems, and prosocial behaviour. In this study, we used a Likert-type response format with three options (0 = not true, 1 = somewhat true, and 2 = certainly true). The validated Spanish version of the SDQ was used in the present study [42]

The Rosenberg Self-esteem Scale (RSS) [43]. This instrument was developed to assess self-esteem. It consists of 10 items scored on a 4-point Likert scale (1 = strongly disagree, 4 = strongly agree). The Spanish version was used in the present study [44].

The Reynolds Adolescent Depression Scale–Short Form (RADS-SF) [45] The RADS-SF is a self-report questionnaire that measures the severity of depressive symptomatology in adolescents. It consists of 10 items using a 4-point Likert scale (1 = almost never, 4 = almost always). The Spanish version adapted and validated for adolescents was used in the present study [46].

The European Bullying Intervention Project Questionnaire (EBIPQ) [47]. The EBIPQ is a self-report questionnaire aimed at measuring traditional bullying and victimisation at school. After a definition of traditional bullying, students were asked to indicate the number of times they have experienced 14 situations (7 for victimisation and 7 for bullying, e.g., “Someone has spread rumours about me”, “Someone has hit me”) during the previous 2 months. Students responded to the 14 items on a 5-point Likert scale (0 = never, 1 = once or twice, 2 = once or twice a month, 3 = once a week, 4 = more than once a week). In this study, we only used items related to victimisation. The psychometric properties of the EBIPQ have been examined previously in Spanish samples [47].

The European Cyberbullying Intervention Project Questionnaire (EBIPQ) [47]. The ECIPQ is a self-report questionnaire that evaluates the dimensions of cyberbullying and cybervictimisation. Following a definition of cyberbullying, students were asked to indicate how many times they have experienced 22 situations (11 for cybervictimisation and 11 for cyberbullying, e.g., “Someone has hacked into my account and pretended to be me”) in the previous 2 months. Again, students responded to the 22 items on a 5-point Likert scale (0 = never, 1 = once or twice, 2 = once or twice a month, 3 = once a week, 3 = more than once a week). In this study, we only used items related to cybervictimisation. The ECIPQ has shown good internal consistency in Spanish samples [47].

The MDS3 School Climate Survey from the Johns Hopkins Center for Youth Violence Prevention [48]. The MDS3 self-report questionnaire was designed to assess a theorised 3-factor model of school climate, which includes safety, engagement, and the environment. The engagement domain includes connection to teachers, student connectedness, academic engagement, school connectedness, equity, and parental engagement. In order to evaluate the participants’ sense of belonging at school, we used items from the connection to teachers, student connectedness, and school connectedness subdomains. The responses were measured using a Likert-type scale with four scores (1 = completely disagree, 4 = completely agree). In the present study, we used the total score, which was the sum of the three subdomains, as an indicator of school engagement. The Spanish adaptation of the MDS3 has demonstrated adequate psychometric properties.

Assessment of academic performance. In order to assess academic achievement, the following question was asked as an indirect estimator: “What was your average grade in the last school year?” with a 5-option Likert-scale format: fail, pass, good, above average, and outstanding.

The Family Affluence Scale-II (FAS-II) [49]. Socioeconomic status was measured using the 4-item child-appropriate measure of family wealth, with scores ranging from 0 to 9. Previous international studies have demonstrated that it has suitable psychometric properties [49].

The Oviedo Infrequency Scale–Revised (INF-OV-R) [50]. The INF-OV-R was administered to the participants to detect those who responded in a random, pseudorandom, or dishonest manner. The INF-OV-R instrument is a self-report questionnaire composed of 10 items in a 5-point Likert-scale format ranging from 1 (completely disagree) to 5 (completely agree). Students with more than two incorrect responses on the INF-OV-R scale were eliminated from the sample. The Spanish version of the INF-OV-R was used in the present study.

### 2.3. Procedure

The tools were administered collectively, in groups of 10 to 30 students, during regular school hours in a classroom that was prepared for this purpose. Administration was supervised by the researchers. The tools were administered by assessors trained in a standard protocol. No incentive was provided for participation. For participants under 18 years old, parents were asked to provide written informed consent for their child to participate in the study. Participants were informed of the confidentiality of their responses and of the voluntary nature of the study. For those adolescents evaluated as being at risk of mental disorders or as having a propensity for suicidal behaviours, resources and support were offered in the form of educational or clinical psychologists.

### 2.4. Data Analyses

First, we calculated the descriptive statistics of all measures. Second, a network of suicide behaviour was estimated. The details of the network analysis have been documented in-depth elsewhere [51,52]. A psychological network consists of nodes (in our case subscales and/or total scores) and edges (unknown statistical relationships between nodes that need to be estimated). We specified a Gaussian Graphical Model (GGM) [53]. This model resulted in conditional dependence relations which are akin to partial correlations: If two nodes are connected in the resulting graph via an edge, they are statistically related after controlling for all other variables in the network. If they are unconnected, they are conditionally independent. For the layout, the Fruchterman–Reingold algorithm was used, placing the strongly connected nodes closer to each other and the least connected nodes further apart. We estimated two inference measures: strength and expected influence. Strength centrality identifies the most important nodes within a network graph. Expected influence is the sum of all edges of a node. To test network stability and accuracy, we used bootstrapping routines implemented in JASP. Since the combination of sample size and number of nodes leads to a considerable computational burden, so far unparalleled in psychological network literature, we performed bootstrap analyses on a high-performance computer cluster. The analyses were performed using SPSS 22.0, JASP, and R.

## 3. Results

### 3.1. Network Structure of Suicidal Behaviour: Links with Protective and Risk Factors

The descriptive statistics for the scores are shown in Table 1. Figure 1 shows the estimated network of suicidal behaviour and related affective, cognitive, and other behavioural psychometric indicators. The estimated network was interconnected, with strong positive edges between suicidal behaviour and risk factors, such as emotional and behavioural problems, and symptoms of depression. Furthermore, suicidal behaviour was negatively connected to self-esteem and personal well-being. In addition, protective factors such as prosocial behaviour, self-esteem, and subjective well-being were more closely associated with each other than with psycho(patho)logical dimensions. In this psychological network, protective and risk factors were inversely related. Figure 2 shows the inference measures of the estimated network. The most central nodes in terms of standardised expected influence were bullying and cyberbullying. The most central node in terms of strength was that of signs of depression.

### 3.2. Network Stability

Stability analysis indicated that the network is accurately estimated, with moderate confidence intervals around the edge weights (see Figure 3 and Figure 4).

## 4. Discussion

The main objective of this study was to analyse the network structure of suicidal behaviour in a large sample of teenagers. Using the Paykel Suicide Scale, we examined the psychological network of suicidal behaviour and its links with tentative risk and protective factors through multiple psychometric indicators. To our knowledge, this is the first study with a representative sample of adolescents aiming to examine the network structure of suicidal behaviour and its relationship with different behavioural and socio-emotional indicators during adolescence. We believe that the traditional insistence on overstating the importance of the diagnostic factor to the detriment of other, equally or more important factors, is an obsolete model that does not respond to the complex, dynamic, contextual, plural, and existential nature of the phenomenon, especially during adolescence and youth. This study attempts to offer a deeper, or at least a different understanding of suicidal behaviour and its links with mental health and emotional well-being and different domains of socio-emotional adjustment. New psychological approaches such as the network model may provide new insights in terms of the boundaries, conceptualisation, understanding, prevention of, and intervention in suicidal behaviour during this critical stage of human development. Our results may be consistent with the conceptual vision that understands suicidal behaviour as a complex network structure of psychological factors (e.g., cognitive, emotional, and behavioural), situations (e.g., bullying at school), and cultural factors (e.g., socio-economic level) that interact with each other over time and levels of analysis.

Analysis of the network topography between suicidal behaviour and the different psychometric indicators (emotional and behavioural difficulties, prosocial behaviour, subjective well-being, self-esteem, depressive symptomatology, academic performance, socio-economic status, school engagement, bullying, and cyberbullying) showed that the most central nodes in that network were depressive symptoms and bullying. The strength centrality index provides specific information regarding the impact of each node on the other nodes in the network. In addition, this study shows that suicidal behaviour was positively related to depressive symptoms, behavioural problems, and bullying, and negatively related to self-esteem and subjective well-being. Furthermore, traditional psychopathology and protective factors (e.g., prosocial behaviour, self-esteem, subjective well-being) were found to be negatively related to variables referring to mental health difficulties (e.g., peer problems, emotional symptoms). It is important to point out that network analysis allows an analysis of the relationship between domains once the effects of all other nodes in the network have been taken into account. This is an interesting aspect, since in the estimated network, for example, the negative association between self-esteem and suicidal behaviour is maintained once the effect of all the other domains is controlled for.

To date, few studies have analysed the suicidal behaviour network using risk and protective factors simultaneously in adolescents, making it difficult to compare our results with previous research [35,54,55,56,57]. For example, a recent study by Beurs et al. [58], using a sample of 3508 young Scottish adults and a battery of psychological tests, found that: (a) internal entrapment was the factor that most contributed to current suicidal ideation, and (b) perceived burden and entrapment were statistically associated with current suicidal ideation, while depressive symptoms were associated with a history of suicidal ideation. Another study with adults [59] found that the highest strength centrality were feelings of depression, hopelessness, perceived burdensomeness, self-esteem, and social support. In particular, self-esteem and social support were shown to be central protective factors. Similarly, self-esteem was also a protective factor in the study by Fonseca et al. [35] with an incidental sample of adolescents. A study by Ordoñez [55] found that suicidal desire is only directly connected to perceived burdensomeness, psychological pain, and defeat. Gijzen et al. [54], using a large community sample of adolescents aged 11–16 years, found that loneliness was a central factor for depression networks and also the most contributing factor of suicide ideation. In general, these studies seem to point towards the importance of examining both protective and risk factors in order to understand the dynamic relationships that are established in estimating the risk of suicidal behaviour or in determining the leap from ideation to suicidal action, for example. The goal is to develop and improve preventive strategies, as well as intervention targets. For instance, the findings from the present study suggest that interventions aimed at decreasing symptoms of depression or increasing self-esteem may be particularly beneficial in reducing suicide risk. In this regard, the implications for the content of school mental health prevention programs are clear [60]. Future research should consider the implications of bullying involvement as a key question in the design of preventive interventions. The results of the present study show that, in the estimated psychological network, one of the most central nodes was bullying. Bullying, cyberbullying, and the peer problems subscales are related, indicating a direct and positive relationship with suicidal behaviour. Social and contextual factors, particularly important during adolescence, have been associated with greater vulnerability to suicide [61]. In fact, involvement in bullying (as a victim, perpetrator, or bully-victim) has been associated with suicidal ideation and behaviour in various meta-analyses [62,63].

Networks highlight complexity between psychological factors and suicide [58]. The study of suicidal behaviour should consider the complex dynamic interaction between biological, psychological, and social factors, experienced by a specific person with a particular history and certain given circumstances. Suicidal behaviour is the end result of the interaction between many different risk factors and cannot be explained by one single factor [58]. The network model fits into this view, since it understands psychological problems as dynamic constellations of symptoms (experiences, traits, etc.) that are causally interrelated, connected through systems of causal relationships. These interactions can occur within the same level of analysis (e.g., at the phenotypic level between symptoms and signs) or between different levels (e.g., between the genetic, brain, cognitive, phenotypic levels). In other words, both horizontally and vertically. In addition, this supposed network of symptoms or mental states can vary over time, from moment to moment. If, for example, a certain relationship of symptoms is activated for a prolonged period of time, it could lead to a psychopathological disorder. Similarly, new interrelationships between symptoms (activation or deactivation) may arise or vary depending on certain conditions of the individual or other circumstances (e.g., environmental conditions, stress, prophylactic interventions, etc.). Ultimately, it is about understanding suicide and suicidal behaviours as dynamic, plural-limit solutions to situations of crisis, rupture, and existential entrapment for which the adolescent cannot find a better solution than to think about ending their life. These behaviours would fulfil a function (not always evident) in this vital biographical context of the adolescent. As we have seen, such behaviours can appear to be associated with a multitude of indicators. It is therefore important to understand that, in essence, they would be the expression of deep suffering, often experienced as intolerable, inescapable, and endless [64].

The present study is not without limitations. First, the use of solely self-reported information limits the conclusions drawn from this work. Second, the study is cross-sectional, so it does not allow the individual dynamic interactions between symptoms to be studied. For that reason, time-series analyses are a logical way forward. Nevertheless, time-series analysis within the field of network analysis is in its early stage, leaving open fundamental questions about how best to estimate a network over time [32]. Third, here, the different psychometric indicators have been considered as tentative risk or protective factors of suicidal behaviour, however, it is also possible that these variables (e.g., depression symptoms) can be seen as another domain of psychopathology. Fourth, the structure of the estimated networks is limited by the tool used. Finally, network analysis is currently in its initial stages and is not exempt from criticism. Although it is shown to be a promising methodology in obtaining important information in a variety of research fields, there are still limitations and issues to be resolved [65,66].

Future studies should incorporate network models with information from multiple levels of analysis (e.g., genetic, brain, cognitive, phenomenological, social, and contextual). In addition, it would be hugely important to collect longitudinal information through new methodologies such as ambulatory assessment to move towards dynamic, contextual, and personalised models.

## 5. Conclusions

Suicidal behaviour can be conceptualised as a dynamic, complex system of cognitive, emotional, and behavioural characteristics. New psychological models allow us to analyse and understand human behaviour from a new perspective. Novel statistical techniques such as network analysis can help us to better study this complexity. This novel conceptualisation as a complex dynamic system is the first step in embracing the complexity of mental health and might help in the identification, prognosis, and prevention strategies for participants at risk for mental difficulties (and other samples). Compared to the other statistical models (e.g., factor-analytic approaches), network analyses provide an informative way to describe the complex relationships between a set of key variables, focusing on the local interactions at the level of smaller units that compose the psychological problems, such as emotional and behavioural manifestations, and not at a disorder level [67]. Network analysis can be seen as a starting point for the move from traditional linear thinking towards a dynamic, contextual model of a complex system. Nevertheless, possibly no model—no matter how comprehensive—can capture or predict all the contingencies that a professional or clinician will have to deal with when faced with the task of treating people at risk of suicide. Consequently, to be effective, we are going to need to be flexible and above all understand why we do what we do and what kind of phenomenon we face. In comparison to the traditional diagnosis-centric approach, the complex, dynamic conception that we have demonstrated here has important practical implications in (1) clinical evaluation, (2) therapeutic aid, and (3) prevention strategies. The way we conceptualise suicidal behaviour directly affects how we help and the relationships we can offer. This is particularly important during adolescence because, by virtue of prevention strategies or the clinical approach, decision-making regarding the support plan will have implications that may be key for adolescents’ future and well-being.

## Figures and Tables

**Figure 1 ijerph-19-01784-f001:**
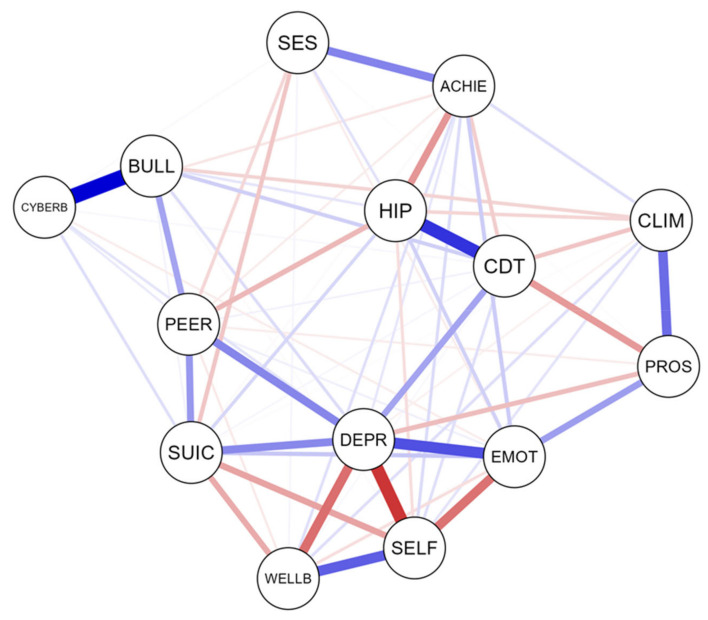
Estimated network of suicidal behaviour, and risk and protective factors. Note: CYBERB = Cyberbullying victimisation; BULL = Bullying victimisation; CLIM = School climate/engagement; SUIC = Suicide behaviour; DEPR = Depression symptoms; PROS = Prosocial behaviour; HIP = Hyperactivity; PEER = Peer problems; CDT = Conduct problems; EMOT = Emotional symptoms; SELF = Self-esteem; WELLB = Personal well-being; ACHIE = School achievement; SES = Socio-economic status. Blue edges represent positive associations, red edges represent negative associations. Thickness and saturation of edges indicate the strength of the associations.

**Figure 2 ijerph-19-01784-f002:**
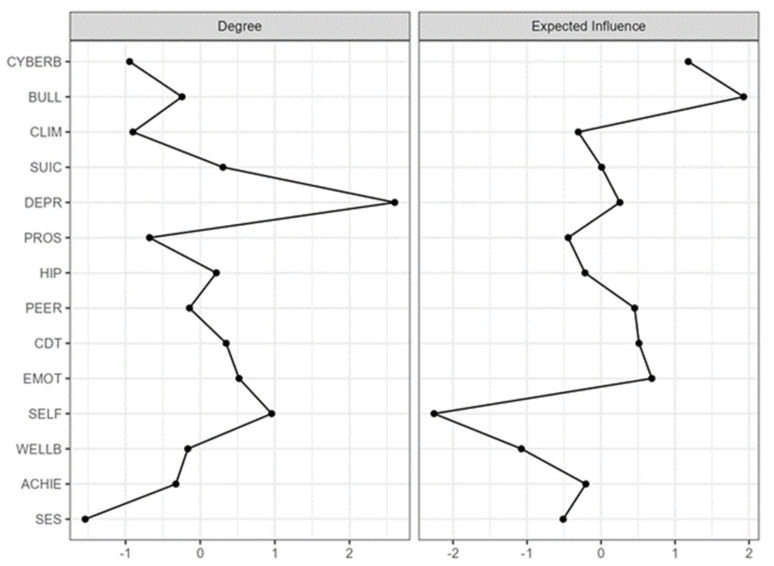
Strength and expected influence values of the suicidal behaviour estimated network. Note: CYBERB = Cyberbullying victimisation; BULL = Bullying victimisation; CLIM = School climate/engagement; SUIC = Suicide behaviour; DEPR = Depression symptoms; PROS = Prosocial behaviour; HIP = Hyperactivity; PEER = Peer problems; CDT = Conduct problems; EMOT = Emotional symptoms; SELF = Self-esteem; WELLB = personal well-being; ACHIE = School achievement; SES = Socio-economic status.

**Figure 3 ijerph-19-01784-f003:**
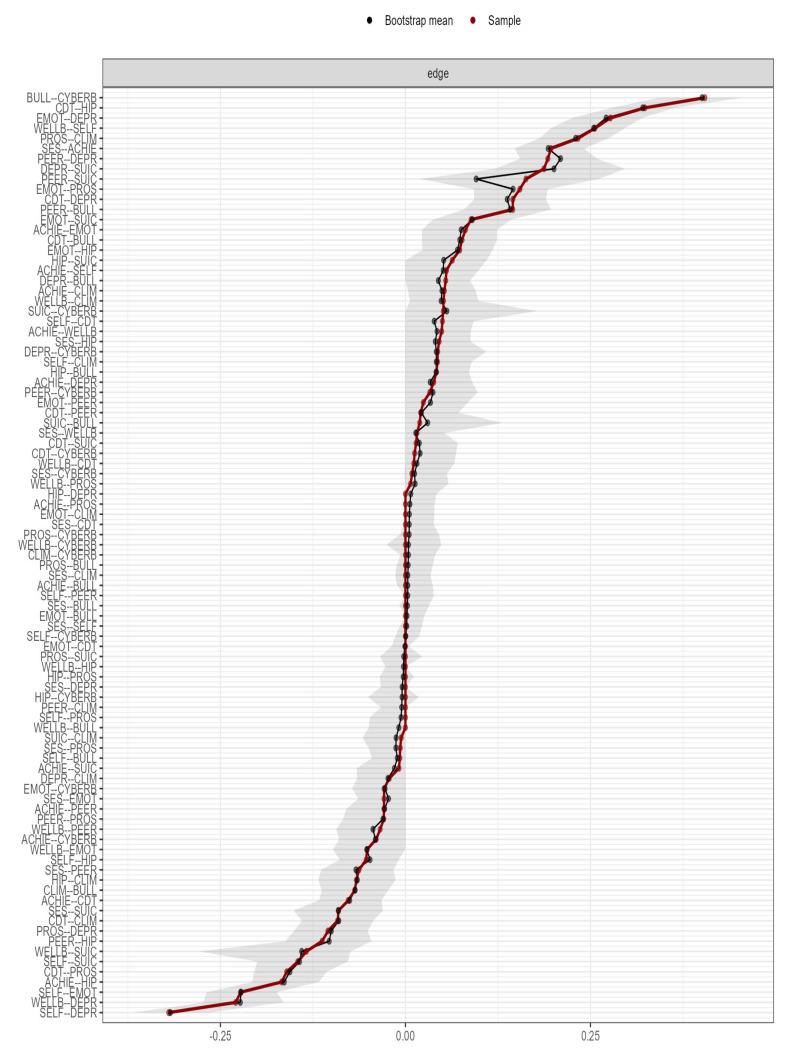
Accuracy of the edgeweight estimates (red line) and the 95% confidence intervals (grey bars) for the estimates.

**Figure 4 ijerph-19-01784-f004:**
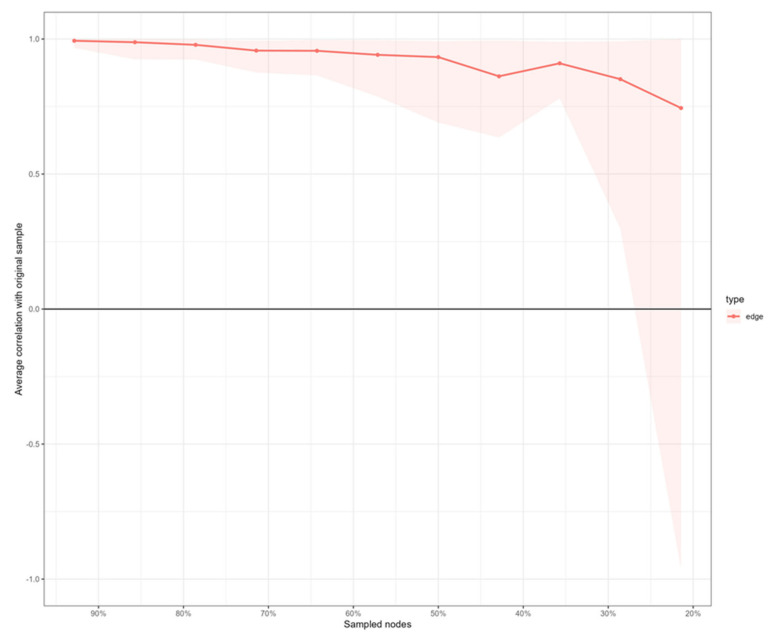
Stability of edges indices of the estimated network.

**Table 1 ijerph-19-01784-t001:** Descriptive statistics for the measures used.

	Mean	*SD*	Skewness	Kurtosis	Minimum	Maximum	Reliability *
Socio-economic status	6.36	1.68	−0.41	−0.28	0	9	0.61
School achievement	6.71	1.76	−0.37	−0.50	3	9.5	**
Personal well-being	7.75	1.86	−1.15	1.80	0	10	**
Self-esteem	30.83	5.56	−0.62	0.31	10	40	0.89
Emotional symptoms	3.44	2.41	0.55	−0.38	0	10	0.75
Conduct problems	1.74	1.55	1.07	1.28	0	8	0.72
Peer problems	1.45	1.59	1.63	3.54	0	10	0.74
Hyperactivity	4.36	2.17	0.07	−0.49	0	10	0.71
Prosocial behaviour	8.56	1.42	−1.16	1.53	2	10	0.78
Depression	16.40	4.49	1.53	3.12	10	40	0.79
Suicide behaviour	0.58	1.12	2.24	4.66	0	5	0.80
School engagement	41.48	6.85	−0.52	1.40	14	56	0.91
Bullying	0.73	1.27	2.13	4.70	0	7	0.83
Cyberbullying	0.20	0.77	6.51	59.93	0	11	0.82

Note. * Mcdonald´s Omega was performed. ** School achievement was measured by 2 items and personal well-being by 1 item.

## Data Availability

The data presented in this study are available upon request from the corresponding author.

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
