# Peer review of "Risk and Protective Factors in Adolescent Suicidal Behaviour: A Network Analysis"

_ijerph, 2022, doi:10.3390/ijerph19031784_

Round 1

Reviewer 1 Report

Despite the fact that this paper is very interesting and a lot of work is visible, some minor issues must be addressed. Please read this paper again - and read carefully. Some spellcheck may be required. For example - in the references section: check on citing nr 57 - I believe that doi number was cut up to "do.". Figures 3 and 4 are unreadable, consider improving their quality or attach them to the supplementary files. About 10 different tools were used in the questionnaires battery, please consider showing their psychometrics and validation studies in one table.  In the discussion section - limitations of the study, the limiting conclusions use of solely self-reported information is mentioned. Please consider describing it possible effects on the conclusions and mirror it as it is a discussion section with other similar studies.

Author Response

1.-Despite the fact that this paper is very interesting and a lot of work is visible, some minor issues must be addressed. Please read this paper again - and read carefully.

We would like to thank the Reviewer for the attention given to our manuscript and for the constructive comments and recommendations. Grammatical and proofreading errors have been checked in the new version of the manuscript.

2.-Some spellcheck may be required. For example - in the references section: check on citing nr 57 - I believe that doi number was cut up to "do.".

Thanks. All references have been checked.

3.-Figures 3 and 4 are unreadable, consider improving their quality or attach them to the supplementary files.

We have added new figures io order to improve their quality.

4.-About 10 different tools were used in the questionnaires battery, please consider showing their psychometrics and validation studies in one table. 

Thanks. We have added this information in table 1.

  1. In the discussion section - limitations of the study, the limiting conclusions use of solely self-reported information is mentioned. Please consider describing it possible effects on the conclusions and mirror it as it is a discussion section with other similar studies.

 Many thanks. In the new version, we have added new limitations of our work. This sentences have been added:

“Second, the study is cross-sectional, so it does not allow the individual dynamic interactions between symptoms to be studied. For that reason, time-series analyses are a logical way forward. Nevertheless, time-series analyses within the field of network analysis is in its early stage, leaving open fundamental questions about how best to estimate a network over time [32]. Third, here the different psychometric indicators have been considered as tentative risk or protective factors of suicidal behaviour, however, it is also possible that these variables (e.g., depression symptoms) can be seen as another domain of psycho-pathology. Fourth, the structure of the estimated networks is limited by the tool used. Finally, network analysis is currently in its initial stages and is not exempt from criticism, although it is shown to be a promising methodology in obtaining important information in a variety of research fields, there are still limitations and issues to be resolved”.

Reviewer 2 Report

This article utilizes a novel method of analysis and includes an impressive combination of risk and protective factors related to adolescent suicidal behaviour. Research studies that incorporate protective factors are certainly needed in the field, and network modeling holds promise for in-depth analysis that linear models might not be able to achieve.

Given that the research was with adolescents and included assessment measures that could necessitate intervention (e.g., recall of suicidal thoughts, depression, bullying) a statement about the provision of resources or availability of supports would be warranted. It is my assumption that approval by the IRB and obtaining parental consent implies that such precautions were taken, but this should be explicitly stated.

For the most part, the description of the statistical modeling was very good, but there is one section that was particularly confusing. From lines 219-220 it seems that the research uses both Strength Centrality and Expected Influence which matches the figures. However, on lines 221-222 it says "We use Expected Influence INSTEAD of strength centrality that has been used in prior work..." (emphasis added). That apparent discrepancy needs to be clarified.

The research findings are interesting, but the end results are not surprising and don't seem especially clinically informative. All the variables seem to have the expected or hypothesized relationships, though the relative strength of certain elements is of interest. Thus, statements referring to prior research, which forms the basis for those expectations, as "obsolete model[s]" (line 276) oversell the novelty of the analysis. If the network analysis was able to provide new insights on how the dynamic relationships work, rather than re-establishing that the relationships exist, it would deserve more recognition for contribution to the field. Use of a more in-depth measure of suicidal experience (i.e., one that is not restricted to a handful of dichotomous items) might contribute to the potential explanatory power of the network model.

Author Response

1.- This article utilizes a novel method of analysis and includes an impressive combination of risk and protective factors related to adolescent suicidal behaviour. Research studies that incorporate protective factors are certainly needed in the field, and network modeling holds promise for in-depth analysis that linear models might not be able to achieve.

Thank you very much for your comments and for giving us the opportunity to respond.

2.-Given that the research was with adolescents and included assessment measures that could necessitate intervention (e.g., recall of suicidal thoughts, depression, bullying) a statement about the provision of resources or availability of supports would be warranted. It is my assumption that approval by the IRB and obtaining parental consent implies that such precautions were taken, but this should be explicitly stated.

            Thanks for this excellent comment. This statement has been added in the new version of our work.

3.- For the most part, the description of the statistical modeling was very good, but there is one section that was particularly confusing. From lines 219-220 it seems that the research uses both Strength Centrality and Expected Influence which matches the figures. However, on lines 221-222 it says "We use Expected Influence INSTEAD of strength centrality that has been used in prior work..." (emphasis added). That apparent discrepancy needs to be clarified.

            We have used both measures of inference. Thanks for this comment.

4.- The research findings are interesting, but the end results are not surprising and don't seem especially clinically informative. All the variables seem to have the expected or hypothesized relationships, though the relative strength of certain elements is of interest. Thus, statements referring to prior research, which forms the basis for those expectations, as "obsolete model[s]" (line 276) oversell the novelty of the analysis. If the network analysis was able to provide new insights on how the dynamic relationships work, rather than re-establishing that the relationships exist, it would deserve more recognition for contribution to the field. Use of a more in-depth measure of suicidal experience (i.e., one that is not restricted to a handful of dichotomous items) might contribute to the potential explanatory power of the network model.

Thanks for this comment. A novel method allowing the investigation of the relationships between different components of psychological processes and difficulties makes use of network analyses. Compared to the other statistical models such as factor-analytic approaches, network analyses provide an informative way to describe the complex relationships between a set of key variables, focusing on the local interactions at the level of smaller units that compose the psychological problems, such as emotional and behaviors manifestations, and not at a disorder level (Borsboom and Cramer 2013; Bringmann and Eronen 2018; McElroy et al. 2018). Furthermore, compared to previous models on psychopathology, the network approach conceptualize the mental disorders as dynamic, complex systems of symptoms and psychological processes interacting in mutually reinforcing loops (Borsboom 2015). From a statistical standpoint, network analyses allow the investigation of the connections (edges) between a series of variables (nodes), introducing novel indices (e.g. centrality indices) that allow the investigation of the role played by each node in the network.